# Obstructive Sleep Apnea and Cardiac Arrhythmias: A Contemporary Review

**DOI:** 10.3390/jcm10173785

**Published:** 2021-08-24

**Authors:** Balint Laczay, Michael D. Faulx

**Affiliations:** Department of Cardiovascular Medicine, Heart, Vascular and Thoracic Institute, Cleveland Clinic, Cleveland, OH 44195, USA; laczayb@ccf.org

**Keywords:** obstructive sleep apnea (OSA), atrial fibrillation (AF), continuous positive airway pressure (CPAP), ventricular tachycardia (VT), atrioventricular (AV) block

## Abstract

Obstructive sleep apnea (OSA) is a highly prevalent disorder with a growing incidence worldwide that closely mirrors the global obesity epidemic. OSA is associated with enormous healthcare costs in addition to significant morbidity and mortality. Much of the morbidity and mortality related to OSA can be attributed to an increased burden of cardiovascular disease, including cardiac rhythm disorders. Awareness of the relationship between OSA and rhythm disorders is variable among physicians, a fact that can influence patient care, since the presence of OSA can influence the incidence, prevalence, and successful treatment of multiple rhythm disorders. Herein, we provide a review of this topic that is intentionally broad in scope, covering the relationship between OSA and rhythm disorders from epidemiology and pathophysiology to diagnosis and management, with a particular focus on the recognition of undiagnosed OSA in the general clinical population and the intimate relationship between OSA and atrial fibrillation.

## 1. Sleep Apnea and Cardiac Rhythm Disorders: An Introduction

Sleep apnea is a highly prevalent disorder among patients with all forms of cardiovascular disease. Decades of data from several large prospective patient registries have revealed that sleep apnea—in particular, obstructive sleep apnea (OSA)—is practically endemic in cardiology clinics and cardiac inpatient wards across the globe [1,2]. OSA has been closely associated with prevalent and incident hypertension [3], ischemic heart disease [4,5], heart failure [6], stroke [7], and all forms of cardiac rhythm disturbance [8]. Additionally, central sleep apnea (CSA) or combined OSA and CSA often affects patients with heart failure and stroke [9]. Sleep apnea and cardiovascular disease are so intertwined with respect to their epidemiology and shared pathophysiology that one can think of them as being two components of a global, multi-system metabolic syndrome driven largely by obesity.

In this review, we will focus on OSA and its relationship to cardiac rhythm disorders. We do this because OSA is the most common form of sleep apnea, and its presence appears to have a greater overall impact on cardiac rhythm disorders than other forms of sleep apnea or sleep-disordered breathing [8]. Additionally, there is a rich and growing body of clinical and basic scientific evidence linking OSA and cardiac rhythm disorders, particularly atrial fibrillation, at multiple levels, which deserves a thorough review [10,11]. Lastly, OSA risk can be readily assessed in the clinical setting by allowing for appropriate testing and subsequent referral for a number of validated treatment options—most commonly, positive airway pressure (PAP) device management. Appropriate treatment can have a positive impact on a patient’s morbidity, mortality, and quality of life, irrespective of its impact on rhythm disorders per se.

## 2. Hiding in Plain Sight? The Epidemiology of Obstructive Sleep Apnea

OSA is a global health crisis that parallels the global obesity epidemic. Obesity and OSA are associated to the extent that it is credible to think of OSA as a consequence of obesity in a majority of cases, although there are certainly patients with OSA who are not obese. In the United States, OSA affects 17% of adult women and 34% of adult men, and incident cases are on the rise [12]. Across the world, OSA prevalence rates vary, but share the trend of rising on every continent [2]. OSA is often associated with features of the metabolic syndrome or “Syndrome X”, including insulin resistance, dyslipidemia, hypertension, and central adiposity, so often so that some authors have proposed the adoption of a “Syndrome Z” to account for the frequent presence of OSA [13]. This association with the metabolic syndrome and its attendant effects on inflammation, oxidative stress, and endothelial dysfunction likely accounts for a large portion of the association between OSA and cardiovascular disease [14].

OSA is often symptomatic, with its principal symptom being excessive daytime sleepiness or fatigue. Historical features that are strongly suggestive of OSA include loud snoring and witnessed apneas or gasping for air during sleep. This element of the history often requires an interview with the patient’s bed partner for confirmation. Other symptomatic manifestations of OSA include difficult concentration, declining work performance, depressed mood, and a heightened risk for motor vehicle accidents. There are a number of valid and simple screening tools that can be easily applied during a patient interview to predict the presence of OSA with fair accuracy. Of these, the STOP-BANG questionnaire (Table 1) appears to have the best sensitivity and specificity for the detection of OSA [15,16,17]. Although many patients with OSA do not volunteer that they are symptomatic, screening for symptoms can nonetheless be helpful. The Epworth sleepiness scale (ESS), an eight-item questionnaire administered during a clinical encounter (Table 2), can prove useful in establishing whether significant OSA symptoms are present [18]. While it is not solely specific to sleepiness caused by sleep apnea, the ESS scale has been well validated in the OSA population and is a reliable gauge for symptom severity. This matters because the presence of subjective and objective sleepiness correlates with greater expression of pro-inflammatory biomarkers and a greater overall risk for adverse cardiac events than the absence of OSA symptoms [19]. The presence of symptoms also justifies OSA treatment, irrespective of any interest in cardiac risk mitigation.

When OSA is strongly suspected after screening, the diagnosis is typically confirmed or excluded with an attended, laboratory-based polysomnogram (PSG) or a home sleep apnea test (HSAT). PSG is considered the gold standard for the diagnosis of sleep disorders owing to its multi-channel data acquisition, which includes brainwave activity and cardiac telemetry to allow for sleep staging, arousal assessment, and assessment of heart rate variability. Major disadvantages of PSG include its limited availability despite rising demand and lack of access in the setting of the ongoing coronavirus disease 2019 (COVID-19) pandemic [20]. In contrast, HSAT offers a simpler dataset that includes continuous oximetry and airflow assessment. The device is worn in the patient’s home and is often more readily available than PSG. HSAT is most appropriate for patients with few medical comorbidities in whom there is a high index of suspicion for OSA, rather than central or mixed apnea. There are even algorithms that allow for the assessment of heart rate variability based on data obtained from continuous oximetry, a feature that may prove useful in the prediction of incident rhythm disorders, such as atrial fibrillation [21].

OSA severity may be assessed in several ways. The most commonly reported metric of OSA severity is the apnea hypopnea index (AHI), which measures the number of times that a patient stops breathing (apnea) or experiences a significant reduction in airflow (hypopnea) per hour of sleep time. An apnea is defined as a lack of an air flow for at least 10 s with an associated oxygen desaturation of at least 4%. Hypopnea is defined as a 50% or greater reduction in airflow for at least 10 s with an associated oxygen desaturation of at least 4%. The AHI is easy to reproduce and is, without question, the most widely reported OSA severity metric in clinical trials. However, the AHI may underrepresent OSA severity when viewed in isolation, and there are data to support focusing more on indices of oxygen desaturation as a gauge of OSA severity [22]. Recent studies have suggested that measures of oxygen desaturation, such as the percentage of sleep time spent with an oxygenation saturation below 90% (T90) or 88% (T88) or the lowest saturation achieved during sleep, may better predict adverse cardiac events than the AHI [23].

## 3. Guilt by Association or Public Enemy Number One? Obstructive Sleep Apnea and Cardiac Arrhythmogenesis

OSA impacts the development of cardiac arrhythmias through direct and indirect mechanisms (Figure 1). The direct effects of OSA on arrhythmia development include the acute physiologic changes that occur as a consequence of airway collapse during sleep, including the development of hypoxemia and hypercapnia [24], changes in sympathetic and parasympathetic tone [25], and fluctuations in thoracic pressure [26]. Indirectly, OSA alters the structure of the heart and is a risk factor for the development of structural heart disease. The indirect effects include the development of cardiovascular disease, including hypertension [3,27], heart failure, and coronary artery disease [28], which form the underlying substrate for arrhythmia development. While atrial fibrillation is the arrhythmia most commonly associated with OSA [29], there is evidence linking OSA to the development of arrhythmias at the level of the sinus node [30], atrial arrhythmias [31], ventricular arrhythmias, and sudden cardiac death [32]. In this section, we will review the pathophysiologic impact of sleep apnea on cardiac arrhythmia and explore the relationship with each disease entity in turn.

The basis of arrhythmogenesis includes changes in myocardial automaticity, triggered activity, and reentrant mechanisms [33]. Abnormal automaticity refers to the formation of cardiac impulses in normally quiescent cardiac cells and is controlled by multiple factors, including sympathetic and parasympathetic tone, acid–base status, and electrolyte disturbances at the membrane and sub-membrane levels [34]. OSA causes repetitive, cyclical changes in sympathetic tone. During apneic events, increased vagal tone causes bradycardia followed by sympathetic discharge as a result of hypoxemia and hypercapnia. Increased vagal tone has been shown to shorten the effective refractory period of the atrium in porcine models of AF and to lead to easier inducibility of AF [35]. The following sympathetic discharge, in turn, promotes increased arrhythmia formation due to beta-adrenergic stimulation [26,36] The repetitive hypoxemia is also thought to increase reactive oxygen species and alter potassium regulation during sleep, which affects the automaticity of cardiac tissue [37]. OSA has also been shown to decrease the atrial effective refractory period (ERP) in canine models, thus leaving the atria more vulnerable to automatic depolarization and ectopy during periods of sleep-disordered breathing [38]. Triggered activity refers to spontaneous depolarizations that are able to cross the membrane potential required to trigger an action potential. These individual extrasystoles can precipitate tachyarrhythmias in both the atrial and ventricular chambers [38]. Well-established causes of triggered activity include hypoxemia, acidemia, and increased sympathetic tone, all of which occur during the repetitive cycles of apnea that characterize OSA. Re-entrant mechanisms are postulated to arise from heterogenous myocardial conduction as a result of abnormal cardiac remodeling in the setting of structural heart disease that accompanies OSA [24].

The mechanistic link between OSA and heart failure is complex and likely bidirectional, with each entity contributing to the other [39]. Obstructive apnea and hypopnea are associated with respiratory efforts against the collapsed upper airway, with associated changes in intrathoracic pressure as high as 60 to 80 mmHg [40] These repetitive, acute swings have a significant impact on cardiac preload and afterload. Simulation of OSA by means of the Mueller maneuver, which involves breathing in against a forced resistance by means of a nose clip and mouthpiece with a 21 G needle, was shown to reproduce changes in intrathoracic pressure in healthy human subjects [38]. This experiment showed that the effect of these intrathoracic pressure swings includes increases in left ventricle (LV) end-systolic volumes, decreased cardiac performance, and abrupt swings in left atrial volumes due to mural stress on the more pliable left atrial wall. Likewise, in patients undergoing cardiac catheterization with measurement of aortic and left ventricular pressures, negative intrathoracic pressures by means of the Mueller maneuver caused increases in LV contraction load as well as an increase in the LV relaxation coefficient (tau) [41,42]. These pathophysiologic changes play a possible role in the observation that severe OSA is associated with ventricular diastolic dysfunction in a dose-dependent fashion [42]. The sum of these interactions is that OSA predisposes one to the development of structural heart disease and heart failure, and the development of these disease states, in turn, predisposes one to and perpetuates the development of OSA.

Sick sinus syndromes, including bradycardia with chronotropic incompetence, sino-atrial exit block, and tachycardia–bradycardia syndromes, are recognized to be more common in OSA patients than in the general population [43,44,45]. One study using Holter monitoring of 239 consecutive patients with a new diagnosis of OSA found that bradyarrhythmias occurred in as many as 20% of the patients and that there was a dose–response effect with respect to oxygen saturation nadir during sleep [46]. Early studies in using tracheostomy as a treatment for OSA in the context of “Pickwickian Syndrome” showed that the treatment of recurrent apneic episodes with tracheostomy normalized both sleep patterns and bradyarrhythmias in this population [47]. These studies were among the first to postulate that hypoxia-induced vagal tone at nighttime could be a significant cause of bradyarrhythmias. A study of patients with excessive daytime sleepiness and sleep-related breathing disorders showed that these patients had increased sympathetic and parasympathetic surges when looking at changes in R-R intervals overnight, showing a link between parasympathetic tone and bradyarrhythmia in this population [48]. A study of six consecutive patients with sleep apnea showed that bradycardia correlated with apneic events and that the duration and severity of bradycardia correlated with the degree of hypoxemia during the apneic events [49] These observations can be explained by the natural diving reflex that is elicited during upper airway obstruction. During upper airway obstruction, there is sympathetic vasoconstriction of arteries to muscles and viscera, with resultant hypertension and vagal tone causing bradycardia [48,50]. This association between OSA and bradycardia is also seen in reverse: Patient cohorts not known to have sleep apnea were shown to have an excessively high prevalence of OSA, regardless of the indication for pacing [51]. Studies of OSA patients referred for pulmonary vein isolation have shown slower sinus node recovery times, suggesting that OSA also impacts the structural integrity of the sinus node [52].

Atrial fibrillation (AF) is the most common arrhythmia in the United States and is estimated to affect more than 3 million individuals [53]. The pathogenesis of atrial fibrillation is complex and incompletely understood, but is accepted to involve both abnormal atrial substrates and triggers of abnormal electrical activity. The initiation of abnormal electrical activity in the pulmonary veins and their subsequent spread and activation of the atrium has been described, and the isolation of said pulmonary veins is the mainstay of catheter-based ablation of atrial fibrillation [54]. In addition to the pulmonary veins, additional areas of abnormal electrical activity have been implicated in AF pathogenesis, including the superior vena cava, the left atrial appendage, the ligament of Marshall, and scarred areas of the left atrium [53]. Additionally, small spiral wave fronts called rotors have been implicated in initiating atrial fibrillation from areas of the atrium outside the traditional pulmonary vein foci [55]. The progression of structural disease, including scarring and fibrosis of the left atrium, has also been implicated in the development and progression of atrial fibrillation [56]. In these patients, the abnormal atrial tissue is considered an additional instigator of atrial fibrillation in addition to the pulmonary veins [57,58]. Parasympathetic tone is also thought to impact the development of atrial fibrillation, with the ganglionated plexi of the left atrium located near the pulmonary vein ostia being an ongoing target of investigation in atrial fibrillation management [59]. The prevalence of OSA is as high as 50–80% in atrial fibrillation patients [59,60,61], and conversely, the prevalence of atrial fibrillation is higher in OSA patients compared to controls (4.8 vs. 1.9%) [44]. OSA predisposes one to the development of atrial fibrillation both through its acute effects in modulating autonomic tone and by acutely changing intrathoracic pressure dynamics, as well as by modulating chronic changes in the underlying atrial substrate [62].

The impact of OSA on structural changes in the left atrium is well described in an increasing body of literature. Studies using mice models of OSA have shown that the repetitive induction of apneic events has direct effects on connexin protein regulation, atrial fibrous tissue content, and structural changes, including slowed atrial conduction [63]. Similar mimics of OSA in rats were shown to selectively increase the fraction of interstitial collagen in the atria of mice, without any similar findings in murine ventricles [64]. This study further showed that Interleukin 6 and Antiogensin-1 Converting Enzyme were significantly upregulated and correlated with the degree of atrial fibrosis [64]. Relating to these laboratory findings, a study of 40 patients undergoing AF ablation showed that while patients with OSA had no differences in baseline AF risk factors compared to controls, they had slower conduction velocities in atrial tissue and more complex electrograms in the atrium [54]. In a study of patients referred for pulmonary vein isolation, 43 patients with OSA were compared to 43 control patients and were shown to have lower atrial voltage amplitude, slower conduction velocity, and more fractionation of electrograms [65].

In addition to the chronic structural changes attributed to OSA, acute changes in physiology account for an additional risk factor for AF development. A retrospective review of overnight polysomnograms from the Sleep Heart Health Study showed that the odds of an arrhythmia were 18 times higher during a period of respiratory disturbance compared to normal breathing during sleep [66]. One acute factor that has been shown to contribute to AF development is hypercapnia. In a study of a sheep model of hypercapnia, there was an increase in vulnerability to the development of atrial fibrillation during the post-hypercapnic phase of airway obstruction [67]. In this experiment, hypercapnia caused a lengthening of the atrial effective refractory period and an increase in conduction time, which resolved with resolution of hypercapnia. Vulnerability to atrial fibrillation development was assessed by evaluating the response to an early electrical stimulus to the atrium, with more development of atrial fibrillation in response to this stimulus during the return to normal carbon dioxide levels.

## 4. Why Will This Patient Not Get Better? The Impact of Obstructive Sleep Apnea on Treatment and Outcomes in Cardiac Rhythm Disorders

As previously mentioned, the prevalence of recognized and unrecognized OSA among patients with cardiac arrhythmias in general and atrial fibrillation in particular is quite high. Thus, screening patients with rhythm disorders for OSA would be reasonable for no other reason than to identify subjects with symptomatic OSA who might benefit from treatment. While altruistic, routine screening of patients with rhythm disorders may also provide insight regarding rhythm management as well, particularly among patients with treatment-resistant rhythm disorders. Obesity and OSA are tightly linked to one another, and both conditions have been recognized as contributors to the reoccurrence of atrial fibrillation after both cardioversion and successful catheter-based ablation [68,69]. The presence of OSA has been associated with a greater rotor burden in patients with atrial fibrillation, with a proclivity for right atrial rotors in particular [70]. OSA has been implicated as a contributor to secondary rhythm-related complications of other cardiovascular diagnoses, including myocardial infarction and heart failure [71,72]. Among patients with permanent pacemakers that are largely implanted for sinus or AV nodal diseases, the prevalence of previously unrecognized OSA is quite high, raising the question of whether appropriate OSA treatment might have resulted in fewer device implants [73]. In patients with non-ischemic cardiomyopathies who have implanted cardiac defibrillators for the primary prevention of sudden death, OSA has been associated with an increased rate of inappropriate shocks [74]. Current clinical guidelines recommend screening all patients with treatment-resistant atrial fibrillation for the presence of OSA, and one should strongly consider screening patients with tachy-brady syndrome or ventricular tachycardia and survivors of sudden cardiac death if OSA risk factors are present [53].

## 5. Do I Really Need to Wear This Mask Every Night? The Impact of Obstructive Sleep Apnea Treatment on Outcomes in Cardiac Rhythm Disorders

Most of the research done to assess the impact of OSA treatment on outcomes in patients with cardiac arrhythmias has focused on PAP devices. To date, there are no published data supporting the use of mandibular advancement device therapy or hypoglossal nerve stimulation for the express purpose of reducing arrhythmic or other cardiovascular events. It has been observed that surgical weight loss can reduce the likelihood for AF recurrence in a dose-dependent fashion, but large-scale randomized clinical trials with rhythm-related endpoints are lacking [75]. While observational cohort data suggest that PAP therapy improves outcomes in rhythm disorders such as atrial fibrillation and lessens the burden of premature ventricular contractions and non-sustained ventricular tachycardia in patients with heart failure [76,77,78,79], recent randomized controlled clinical trials involving subjects with OSA treated with PAP or either sham PAP or no PAP have failed to demonstrate any significant benefits in patients with atrial fibrillation or other arrhythmias [80,81,82]. The reason for this disconnect between observational and randomized trial data is likely multifactorial, but patient adherence to therapy selection probably plays a large role. Current randomized controlled trials involving PAP tend to enroll asymptomatic or minimally sleepy patients due to ethical concerns about not treating sleep patients with PAP. These are also the patients who are less likely to adhere to PAP therapy. Many of these studies also exclude patients with more extreme obesity. There are data linking OSA symptoms to greater OSA morbidity and mortality, so by excluding these patients from clinical trials, we may be testing a lower-risk population than that seen in everyday clinical practice [83,84]. Thus, current randomized clinical trials are likely excluding patients who would be expected to benefit the most from PAP therapy.

## 6. Where Do We Go from Here? Parting Thoughts and Future Directions

Based on a review of the available data, it seems clear that the presence of OSA increases one’s likelihood for developing incident atrial fibrillation, nocturnal pauses, bradycardia, sustained and non-sustained ventricular arrhythmias, and individual ectopic ventricular complexes. The growing body of basic scientific data supporting the causal role of OSA-related events in the genesis of rhythm disorders is quite robust. Observational data also strongly suggest that the presence of unrecognized and untreated OSA interferes with the success of conventional rhythm management, especially in patients with AF. What has yet to be clearly established is whether OSA treatment—and PAP treatment in particular—actually improves rhythm-related outcomes in patients with OSA. It is well known that the presence of objective and subjective sleepiness in OSA is associated with poorer cardiac outcomes for reasons that are not entirely clear, but may be related to a greater degree of oxidative stress and the expression of pro-inflammatory molecules in these sleepy patients [83,84] While randomized clinical trial data looking at the effect of PAP treatment on cardiac outcomes have been admittedly disappointing [80,85], these trials enrolled patients with few or no OSA symptoms for ethical reasons, and many of these trials studied patients who were much less obese than the average “real-world” OSA patient. PAP compliance also remains a limitation in many clinical trials [86]. These facts raise significant doubts about the true efficacy of PAP treatment in such patients, and future trials should look to include sleepy patients with higher BMIs to see if this lack of treatment effects persists. Since recent randomized clinical trials have called into question whether PAP therapy provides any cardiovascular therapy at all, it may be time to revisit the ethics of randomizing sleepy patients in PAP trials or to utilize different study designs to address these questions, such as observational studies using propensity scoring [86]. In addition, the role of a multi-faceted intervention for OSA, such as combining PAP with structured weight loss, exercise, and lifestyle and nutritional counseling, deserves more exploration, as there are data that suggest that these approaches may benefit patients with rhythm disorders more than PAP therapy alone [87].

## Figures and Tables

**Figure 1 jcm-10-03785-f001:**
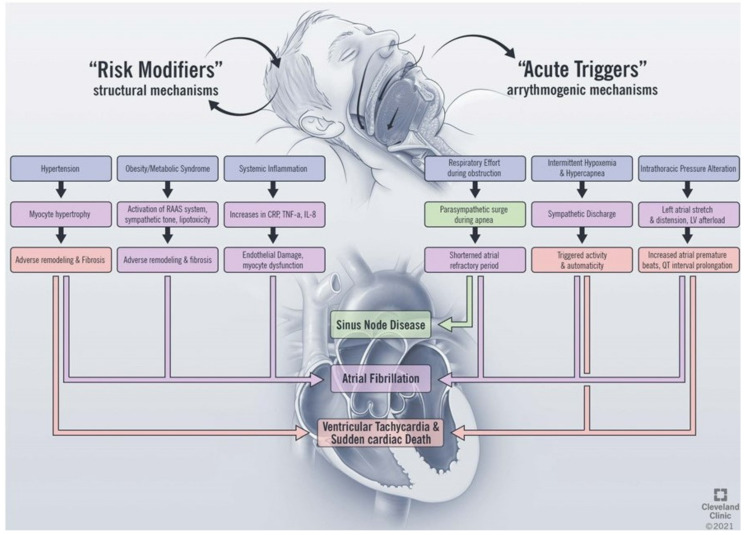
Proposed mechanisms linking obstructive sleep apnea and cardiac arrhythmias. RAAS, renin angiotensin aldosterone system; CRP, c-reactive protein; TNF, tumor necrosis factor; IL, interleukin; LV, left ventricle.

**Table 1 jcm-10-03785-t001:** The STOP-BANG questionnaire and its accuracy in detecting moderate or severe sleep apnea (AHI ≥15/hour). Score one point for each finding.

Snoring		Typically loud and disruptive		
Tiredness		Tired, fatigued, or sleepy during the day		
Observed apnea		Often observed by bed partner		
Pressure		History of hypertension treatment		
BMI		BMI > 35 kg/m^2^		
Age		>50 years		
Neck circumference		>40 cm		
Gender		Male		
**STOP BANG Score**	**Sensitivity**	**Specificity**	**PPV**	**NPV**
1	100	1	67	100
2	99	10	68	79
3	94	32	73	74
4	81	51	76	58
5	60	72	80	48
6	35	89	86	42
7	14	96	88	37
8	3	100	95	35

AHI, apnea–hypopnea index; BMI, body mass index, PPV, positive predictive value; NPV, negative predictive value. The table was created from data taken from references [17] and [19].

**Table 2 jcm-10-03785-t002:** The Epworth Sleepiness Scale (ESS) and its relationship to OSAS risk. Each question is scored from 0 to 3. ESS score range is from 0–24.

Activity		Likelihood of Dozing 0 = Never, 1 = Slight, 2 = Moderate, 3 = High	
Sitting and reading			
Watching television			
Sitting inactively in a public place			
As a car passenger for one uninterrupted hour			
Lying down in the afternoon when able			
Sitting and talking to someone			
Sitting quietly after lunch with no alcohol			
In a car, while stopped for a few minutes in traffic			
**Mean RDI**	**Mean ESS**	**ESS Range**	**Interpretation**
8.8 ± 2.3	9.5 ± 3.3	1–9	No or little OSAS risk
21.1 ± 4.0	11.5 ± 4.2	10–15	Moderate OSAS risk
49.5 ± 9.6	16.0 ± 4.4	16–24	High OSAS risk

RDI, respiratory disturbance index; OSAS, obstructive sleep apnea syndrome. The table was adapted from reference [20].

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
