# Peer review of "Obstructive Sleep Apnea and Cardiac Arrhythmias: A Contemporary Review"

_jcm, 2021, doi:10.3390/jcm10173785_

Round 1

Reviewer 1 Report

I commend the authors on this excellent review article. In terms of flow, the manuscript is well structured with appropriate transitions to different topics. The authors also do an excellent job describing the global aspect of OSA on the heart with a description ranging from the tissue level to the systems level. Furthermore, I appreciate the clinical impact the authors portray. This review is not only helpful to the cardiologist and pulmonologist, but is helpful to the primary care physician who sees these patients on a regular basis during initial evaluation. 

Author Response

We thank the reviewer for the kind and complimentary feedback.  We drafted this manuscript in a fashion that we hoped would be applicable to a broad audience including cardiologists, pulmonologists and internists so we are pleased that the reviewer feels that our prose is appropriate for the target audience.

Reviewer 2 Report

Dear Authors, the paper is simple and easy to read.

The Q&A form is interesting.

I approve the publishing this article  in the hope of sensitizing cardiologists to OSAS pathology.

Some corrections to do:

In the United States OSA affects 17% of adult women and 34% of adult men and 38 incident cases are on the rise [13]. PLEASE CITE THE ORIGINAL PAPER, DON'T THE REVIEW

CORRECT via direct and indirect mechanisms IN THE FORM through direct and indirect mechanisms

CORRECT THIS GRAMMATICALLY INCORRECTED SENTENCE Most of the research done to assess the impact of OSA

Author Response

We thank the reviewer for the thoughtful and constructive feedback. Below please find our itemized responses to your suggestions:

In the United States OSA affects 17% of adult women and 34% of adult men and 38 incident cases are on the rise [13]. PLEASE CITE THE ORIGINAL PAPER, DON'T THE REVIEW

We appreciate the attention to this oversight.  The original article by Peppard et al has been appropriately cited.

CORRECT via direct and indirect mechanisms IN THE FORM through direct and indirect mechanisms

The sentence now reads "...through direct and indirect mechanisms..."

CORRECT THIS GRAMMATICALLY INCORRECTED SENTENCE Most of the research done to assess the impact of OSA

Thank you.  The revised sentence reads as follows:  "Most of the research done to assess the impact of OSA treatment on outcomes in patients with cardiac arrhythmias has focused on PAP devices".